# Hydrogel-Mediated DOX⋅HCl/PTX Delivery System for Breast Cancer Therapy

**DOI:** 10.3390/ijms20194671

**Published:** 2019-09-20

**Authors:** Hoon Hyun, Young Bum Yoo, So Yeon Kim, Hyun Sun Ko, Heung Jae Chun, Dae Hyeok Yang

**Affiliations:** 1Department of Biomedical Sciences, Chonnam National University Medical School, Gwangju 61469, Korea; hhyun@chonnam.ac.kr; 2Department of Surgery, School of Medicine, Konkuk University, Seoul 05030, Korea; 0117652771@kuh.ac.kr; 3Department of Dental Hygiene, College of Health Sciences, Cheongju University, Cheongju 28503, Korea; goodany00@gmail.com; 4Department of Obstetrics & Gynecology, College of Medicine, The Catholic University of Korea, Seoul 06591, Korea; mongkoko@catholic.ac.kr; 5Department of Medical Life Sciences, College of Medicine, The Catholic University of Korea, Seoul 06591, Korea; chunhj@catholic.ac.kr; 6Institute of Cell and Tissue Engineering, College of Medicine, The Catholic University of Korea, Seoul 06591, Korea

**Keywords:** injectable glycol chitosan, doxorubicin hydrochloride, beta-cyclodextrin, paclitaxel, breast cancer therapy

## Abstract

We used a hydrogel-mediated dual drug delivery approach, based on an injectable glycol chitosan (GC) hydrogel, doxorubicin hydrochloride (DOX⋅HCl), and a complex of beta-cyclodextrin (β-CD) and paclitaxel (PTX) (GDCP) for breast cancer therapy in vitro and in vivo. The hydrogel was swollen over 3 days and remained so thereafter. After an initial burst period of 7 hours, the two drugs were released in a sustained manner for 7 days. The in vitro cell viability test showed that GDCP had a better anticancer effect than well plate and DOX⋅HCl/PTX (DP). In addition, the in vivo tests, which evaluated the anticancer effect, systemic toxicity, and histology, proved the feasibility of GDCP as a clinical therapy for breast cancer.

## 1. Introduction

Breast cancer is one of the most common cancers in women worldwide. Owing to the presence of lymph nodes next to the breast, metastasis of breast cancer often occurs in younger people with active cell proliferation [1]. Most early-stage breast cancers are removed by surgery; however, if the lesions are more severe, chemotherapy is first administered to decrease tumor size. However, most known chemotherapies are accompanied by side effects, such as vomiting, anorexia, and hair loss, because the drugs affect both cancer cells and normal cells [2].

To overcome these problems, there have been many studies of systemic drug delivery systems using intravenous injection of nanoparticles [3]. However, these nanoparticles face environmental changes, such as changes in pH and salt conditions, and come in contact with various plasma proteins in the blood stream; these environmental factors destabilize nanoparticles through changes in the thermodynamic or kinetic equilibrium [4]. Therefore, given these obstacles, local drug delivery systems are a good candidate to overcome the drawbacks of systemic drug delivery systems [5]. In our previous studies, visible-light-cured glycol chitosan (GC) hydrogel was prepared as a local drug delivery system, and this platform was found to have potential for several solid cancer therapies [6,7,8].

In addition to systemic drug delivery complications, another obstacle to breast cancer chemotherapy is multidrug resistance (MDR) [9,10]. Combination chemotherapy is known to avoid MDR in breast cancer [11]. Paclitaxel (PTX), a type of taxane, is an antimicrotubulin agent with anticancer effects against breast cancer [12]. It is known that PTX promotes tubulin dimerization and inhibits the depolymerization of microtubules, which enhances the anticancer effect [12]. Doxorubicin (DOX) is an anthracycline chemotherapy drug and a useful drug for breast cancer therapy [13]. DOX damages the DNA of cancer cells and the damaged DNA blocks the division of cancer cells, eventually killing the cells [14].

Herein, we, as shown in Table 1, prepared three kinds of samples for this study and described the preparation of a local drug delivery system consisting of visible-light-cured GC hydrogel and a combination of chemotherapy PTX and doxorubicin hydrochloride (DOX⋅HCl) in water-soluble forms for the treatment of advanced breast cancer. Beta-cyclodextrin (β-CD) was used as an excipient to dissolve PTX in water because it can improve the water solubility of the drug via the formation of an inclusion complex [8]. The therapeutic efficacy of the newly designed drug delivery system in breast cancer was investigated in vitro and in vivo.

## 2. Results

### 2.1. Preparation and Swelling Ratio of DOX⋅HCl/PTX-loaded GC Hydrogel (GDCP)

The swelling ratio of the sample was measured every day for 7 days at 37 °C and pH 7.4, as shown in Figure 1. The swelling ratio increased for 3 days and remained constant thereafter. It is thought that the volume of the sample increased as the entangled polymer chains were stretched by the penetration of water molecules across the cross-linked network; subsequently, the stretched polymer chains were no longer expandable by crosslinking, which contributed to the maintenance of the constant swelling ratio.

### 2.2. Release Behavior of DOX⋅HCl and CD/PTX

The release behaviors of DOX⋅HCl and CD/PTX from GDCP in phosphate buffered saline (PBS) (pH 7.4) at 37 °C for 7 days were compared with those from GC/DOX⋅HCl/PTX (GDP) and are shown in Figure 2. The release profile of DOX⋅HCl was similar throughout the period tested (Figure 2A). This was confirmed by the improvement in the water solubility of the salt form of DOX. The DOX⋅HCl showed rapid release behavior for 7 hours, and sustained release behavior thereafter. This was attributed to the location of the drugs, as the molecules located on the hydrogel surface were rapidly released and the molecules in the hydrogel core were released slowly by diffusion. In the case of PTX, the two hydrogels exhibited different release behaviors, because water-soluble CD/PTX should be easily migrated from the matrix (Figure 2B).

### 2.3. In Vitro Anticancer Effect

The effect of GDCP on the in vitro cell proliferation rate of MCF-7 cells was measured after culture for 1, 3, 5, and 7 days to investigate the anticancer effect of the sample (Figure 3). The cell proliferation rate was compared with cells cultured in a 96-well plate (control) and treated with DP. In addition, the cell proliferation rate was evaluated in the surface and bulk of GDCP. Compared with the control, the cell proliferation rate of the drug-treated cells gradually decreased over time. The cell proliferation rates in the surface and bulk of GDCP were lower than those of DP. In particular, the cell proliferation rate in the bulk was lower than that in the surface.

### 2.4. In Vivo Anticancer Effect

The anticancer effect of GDCP in mice injected with MCF-7 cells is shown in Figure 4. As shown in Figure 4A, the gross appearance of tumors treated with DP, GDCP, and the control was observed 1, 3, 5, and 7 days after the injection of the test samples. In the control, a gradual increase in the tumor size was observed throughout the period. Both of locally injected DP and systemically injected GDCP exhibited the gradual decrease of tumor size as a function of time. In particular, the GDCP exhibited greater anticancer effect than the DP. The appearances of tumors resected on Day 7 are shown in Figure 4B. As shown in Figure 4B, the cancers treated with the samples were smaller than those treated with the control. In addition, GDCP resulted in a marked decrease in tumor size compared with DP. The sizes of the tumors in the control and samples were measured at the indicated time points (Figure 4C). On Day 0, the average sizes of the control, DP-treated, and GDCP-treated tumors were 173, 176, and 178 mm^3^, respectively. In the control group, the tumor size increased gradually for 7 days and reached 356 mm^3^. DP slightly decreased tumor size over 7 days, to 131 mm^3^. As expected, GDCP significantly decreased the tumor size over 7 days, to 60 mm^3^.

### 2.5. Systemic Toxicity

The body weight of mice in each group was examined to evaluate the toxicity of the tested samples for 7 days after injection on Day 0 (Figure 5). A gradual increase in the body weight was observed in the control due to the growth of the tumor. In contrast, treatment with DP and GDCP resulted in a decrease in body weight. GDCP induced a noticeable decrease in body weight. This result was attributed to the anticancer effect of GDCP.

### 2.6. Cardiotoxicity

The histological changes in the hearts of mice treated with DOX⋅HCl/PTX and GDCP for 7 days compared with the hearts of mice in the control group are shown in Figure 6. In the control group, normal histology was observed; in contrast, some abnormal cardiotoxicities, such as disorganization of the myocardium and myofibrillar fragmentation, were observed in the hearts of mice in the DOX⋅HCl/PTX-treated group. Meanwhile, normal histology was observed in the GDCP-treated group. These results suggested that the local injection of the GC hydrogel was an efficient anticancer delivery system.

### 2.7. Histological Evaluations

The histological changes in cancer cells extracted from the GDCP-treated group were compared with those in cancer cells extracted from control and DP-treated groups (Figure 7). In the control group, the cancer cells were densely distributed. In the DP-treated group, not only cancer cells, but also partially necrotic tissues, were observed. Overall, necrotic tissues were observed in the GDCP-treated group. The results indicated the efficacy of the local GC system for anticancer drug delivery.

## 3. Discussion

Conventional drug administration, such as the systemic drug delivery system, often requires repeated administration at a high dosage to provide a sufficient therapeutic effect; however, this may result in low efficiency, poor patient compliance, and serious side effects [15]. In our previous study, we designed and prepared a local drug delivery system that used a visible-light-cured GC hydrogel to supplement the drawbacks of a systemic drug delivery system [6,7,8]. The hydrogel system was found to be a good platform for the delivery of anticancer drugs to target sites for improved anticancer effects [6,7,8].

However, the hydrogel system may not have sufficient potential for the chemotherapeutic effects of breast cancer, because many chemotherapy drugs, such as taxol, anthracyclines, mitoxantrone, topotecan, and etoposides, induce MDR, which is a known major factor in the failure of cancer treatment [16,17]. The combination of anthracyclines and taxanes has been shown to be effective for breast cancer treatment [18,19]. In particular, the main focus of several research groups has been the investigation of the effect of the combination of DOX and PTX, as representatives of anthracyclines and taxanes, on breast cancer treatment [18,19].

Another consideration is that most anticancer drugs are poorly water soluble, which has limited their clinical use. DOX is commonly used in the salt form (DOX⋅HCl) to improve water solubility. As reported previously, we used β-CD as an excipient to improve the water solubility of PTX [8]. β-CD can improve the water solubility of anticancer drugs owing to its unique structure, which leads to inclusion complexes with the drugs followed by endowing water solubility [8].

Based on these considerations, we prepared a local drug delivery system of GDCP, consisting of visible-light-cured GC hydrogel, DOX⋅HCl, and the inclusion complex between β-CD and PTX, for breast cancer treatment, and investigated the release behavior of DOX⋅HCl and PTX from GDCP compared with that from GDP. The results demonstrated that GDCP resulted in more rapid release of DOX⋅HCl and PTX than GDP owing to the effects of β-CD and swollen GC hydrogel in aqueous solution (Figure 1 and Figure 2). Owing to the three-dimensional network of the hydrogel, it becomes swollen in aqueous conditions by the invasion of water molecules into the open spaces (pores), which results in increased pore sizes [20]. Pore size affects the diffusion-controlled release of drugs [20]; drugs that are smaller than the pore size can migrate freely [20]. In addition, β-CD improves the water solubility of PTX through the formation of an inclusion complex, and solubilized drug molecules can be migrated by diffusion. 

To establish a local drug delivery system, GDCP was injected to normal tissues near the cancer, and the drugs moved toward the cancer. This administration indicated that DOX⋅HCl and CD/PTX were released at physiological pH. At pH 7.4, DOX⋅HCl and CD/PTX showed similar release behavior. This demonstrated that the hydrogel system has a sufficient pore size to enable the release of DOX⋅HCl and CD/PTX by diffusion.

PTX is a crystalline compound that is highly insoluble in water owing to its hydrophobic nature [21]. It exerts anticancer activity by inhibiting microtubule formation and inducing cell cycle arrest [21]. DOX can damage DNA by intercalation, metal ion chelation, and the generation of free radicals [14]. In addition, it interferes with the action of DNA topoisomerase II, an important enzyme for DNA function, which induces apoptosis in cancer cells [14]. One of the most serious side effects of DOX is cardiotoxicity, arising from drug-induced cardiomyopathy, which has a poor prognosis and is sometimes fatal [22,23,24]. Moreover, the present available treatments do not result in improvements in cardiomyopathy. Currently, systemic and local drug delivery strategies, such as nanomaterial-based systems with passive/active targeting, liposome-based systems, and hydrogel systems, have been reported [25,26,27].

As expected, the codelivery of DOX⋅HCl and PTX using a GC hydrogel for local drug delivery and β-CD for increased water solubility of PTX yielded better in vitro and in vivo anticancer effects against breast cancer compared with the systemic codelivery system of DOX⋅HCl and PTX (Figure 3, Figure 4, Figure 5, Figure 6 and Figure 7). Furthermore, local administration resulted in a lower occurrence of cardiomyopathy than systemic administration (Figure 6). Therefore, we suggest that GDCP may be a potential clinical therapy for breast cancer (Figure 8).

## 4. Materials and Methods

### 4.1. Materials

GC (≥60% calculated by titration, crystalline, MW ≈ 585,000 g/mol) and glycidyl methacrylate (GM) for visible-light-cured hydrogel preparation were purchased from Sigma-Aldrich (St. Louis, MO, USA). Riboflavin 5′-monophosphate sodium salt (riboflavin) used for photo curing was supplied by Santa Cruz Biotechnology, Inc. (Santa Cruz, CA, USA). Doxorubicin hydrochloride (DOX⋅HCl; Tokyo Chemical Industry Co., Ltd, Tokyo, Japan) and paclitaxel (PTX; Shin Poong Pharm, Co., Ltd.; Ansan, Kyunggi, Republic of Korea) were used for breast cancer therapy. β-CD (Sigma-Aldrich; St. Louis, MO, USA) was used to improve the water solubility of PTX. The cellulose membrane was purchased from Spectrum Laboratories Inc. (Rancho Dominguez, CA, USA) and theMCF-7 human breast cancer cell line was obtained from American Type Culture Collection (ATCC; Manassas, VA, USA). Cell counting kit-8 (CCK-8; Dojindo Molecular Technologies, Inc. Rockville, MD, USA) was used for the in vitro cell viability assay. All chemicals were used as received.

### 4.2. β-CD and PTX Complex (CD/PTX)

This complex was prepared as reported previously [8]. β-CD (2 mmol, 3 mg) and PTX (2 mmol, 1 mg) were dissolved in water (5 mL) and acetone (5 mL), respectively. The PTX solution was added dropwise into the β-CD solution and stirred for 48 h continuously. The filtered solution was lyophilized and stored in a desiccator at −20 °C before use.

### 4.3. Preparation of Methacrylated GC (GM-GC)

GM-GC was prepared as reported by our studies previously [6,7,8]. To a solution of GC (0.003 mmol, 1.5 g) in water (500 mL), GM (0.05 mmol, 7 mg) was added and adjusted to pH 9 with continuous agitation. After stirring for 2 days, the solution was neutralized and dialyzed (membrane with a molecular weight cut-off: 20 kDa) in water for 7 days. The final white product was obtained by lyophilization at −90 °C for 7 days before use. 

### 4.4. Preparation of Injectable GDCP Hydrogel

Riboflavin (12 µM), DOX⋅HCl (2 mg), and CD/PTX (PTX: 2 mg) were added in GM-GC hydrogel precursor solution and adjusted to a final concentration of 1 w/v%. After mixing to ensure homogeneity, the solution was irradiated for 10 s by using blue light (430–485 nm, 2,100 mW/cm^2^, light-emitting diode curing light, FoshanKeyuan Medical Equipment Co., Ltd., Guangdong, China) to induce hydrogel formation. The swelling ratio of the hydrogel was calculated as the ratio of the swollen weight to the initial weight. The swollen weights were measured at 37 °C for 7 days. 

### 4.5. Release Test of DOX⋅HCl and PTX in GDP and GDCP

GDP and GDCP (DOX⋅HCl: 2 mg/mL and PTX: 2 mg/mL) were added to cellulose membrane tubes (MWCO: 3,500 g/mol) separately, and the tubes were immersed in conical tubes filled with 8 mL PBS (pH 7.4). During continuous incubation at 37 °C with agitation at 100 rpm, efflux (2 mL) was extracted at the indicated time points (1, 3, 6, 12, 24, 48, 72, 96, 120, 144, and 168 h) and an equal volume of fresh PBS was added. The release of DOX⋅HCl and PTX was detected by using a UV-visible spectrophotometer at 480 nm and high-performance liquid chromatography (HPLC) with a mobile phase composed of acetonitrile and water (50:50, v/v%) at 227 nm, respectively. The HPLC was equipped with a UV detector (1100 series, Agilent Technologies, Palo Alto, CA, USA) and an Ascentis C18 column (25 cm × 4.6 mm, particle size: 5 µm; Supelco, St. Louis, MO, USA).

### 4.6. In Vitro Cell Prolferation Rate Measurement

Cell proliferation rate was evaluated by using a CCK-8 assay. The GDCP (DOX⋅HCl: 44 μg/mL and PTX: 44 μg/mL corresponding to the injection concentration used in in vivo animal test) hydrogel precursor solution dissolved in water for injection was layered on a Transwell plate. After visible light irradiation for 10 s, MCF-7 cells (5 × 10^3^ cells/well) were seeded on the surface of the hydrogel, and then incubated at 37 °C for 1, 3, 5, and 7 days. The control treatment was a combination of DOX⋅HCl (44 μg/mL) and PTX (44 μg/mL). The mixture was added in culture media (1 mL) consisting of DMEM supplemented with 10% fetal bovine serum and 1% penicillin/streptomycin, and sonicated for 3 minutes. The sonicated drug solution was spread on the cell-attached plate (5 × 10^3^ cells/well) and incubated at 37 °C for 1, 3, 5, and 7 days. At the indicated time points, the cell-seeded hydrogel and the drug-treated cells were washed with PBS, and CCK-8 (10 μL) was added to the plates. After incubation for 4 h, the absorbance was measured at 450 nm.

### 4.7. Establishment of MCF-7 Tumor-Bearing Mouse Model

The in vivo animal experiments were conducted in accordance with the protocols approved by Chonnam National University Animal Research Committee (Approval Number: CNU IACUC-H-2017-64). Adult male NCRNU nude mice (six weeks old, 22–25 g, *n* = 6 per group and time point; Seongnam, Republic of Korea) were used for the preparation of an MCF-7 tumor-bearing mouse model. The cells (5 × 10^6^ cells) were suspended in water for injection (100 µL; Jeil Pharmaceutical Co. Ltd., Daegu, Republic of Korea). The suspension was subcutaneously injected into the back of each mouse and the animals were cared for and monitored until the tumor reached approximately 1 cm.

### 4.8. Administration of DP and GDCP

DOX⋅HCl (2 mg/kg) and PTX (2 mg/kg) were dissolved in 100 μL of water for injection containing 0.01% DMSO. This solution used as a control was administered once per day for 7 days to the cancer-bearing mice via the tail vein (*n* = 6). For injecting photo-cured GDCP, GM-GC hydrogel precursor solution (1 w/v%) containing (DOX⋅HCl: 2 mg/kg and PTX: 2 mg/kg) was first irradiated for 10 s to authorize hydrogel functionality. Afterward, the GDCP (DOX⋅HCl: 2 mg/kg and PTX: 2 mg/kg) was subcutaneously injected close to the cancer. The change in cancer volume and body weight of mice administered with each treatment was measured once every 2 days after administration. The cancer volume was calculated from the following formula: V = 0.5 × longest diameter × (shortest diameter)^2^. Body weights at determined time intervals were measured. 

### 4.9. Histological Evaluation

Histological evaluation was carried out by two observers from different agencies. All sample mice were sacrificed on Day 7. Then, the cancer and heart tissues extracted from cancer-bearing (control), systemically DP-treated, and locally GDCP-treated mice were stained with hematoxylin and eosin (H&E) to investigate in vivo anticancer effects and cardiotoxicity, respectively. In the cardiotoxicity assay, heart tissue in a normal mouse was extracted as a control. All tissues were fixed in 4% (v/v) formaldehyde, and dehydrated in graded ethanol series (100%, 95%, 80%, and deionized water). After embedding in paraffin, the blocked tissue was sliced to obtain 3 μm thick sections. The sections were stained with H&E according to a general staining protocol. After the staining procedures, a drop of Permount was placed on each slide and was allowed to spread beneath the coverslip. The stained slides were observed using slide scanner (Pannoramic MIDI; 3DHISTECH Ltd., Budapest, Hungary) and panoramic viewer (Version 1.15.3; Pannoramic MIDI; 3DHISTECH Ltd., Budapest, Hungary) program.

### 4.10. Statistical Analysis

Data of in vitro cell viability, cancer volume, and body weight were expressed as the mean ± standard deviation. Statistical analysis was computed by one-way analysis of variance (ANOVA) using SPSS software (SPSS Inc., Chicago, USA). A value of ^#^*p* < 0.05 was considered statistically significant.

## 5. Conclusions

In this study, we investigated the feasibility of a DOX⋅HCl and PTX-loaded visible-light-cured GC hydrogel system for breast cancer therapy. Both water-soluble DOX⋅HCl and β-CD complexed PTX were released in a sustained manner for 7 days, together with an initial burst release of 7 h. The in vitro cell viability test exhibited a superior anticancer effect of GDCP. In addition, the in vivo tests, animal experiments, and histological evaluations revealed that GDCP resulted in a remarkable decrease in cancer size and negligible cardiotoxicity. Therefore, we concluded that GDCP is a potential clinical therapy for breast cancer.

## Figures and Tables

**Figure 1 ijms-20-04671-f001:**
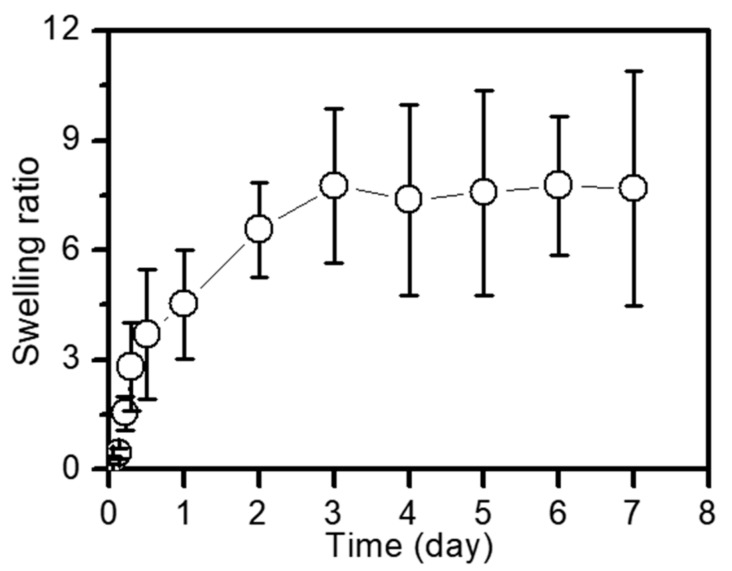
Swelling ratio of GDCP hydrogel measured at day 0, 1, 2, 3, 4, 5, 6, and 7. This experiment was carried out three times at 37 °C. These values were expressed as the mean ± standard deviation.

**Figure 2 ijms-20-04671-f002:**
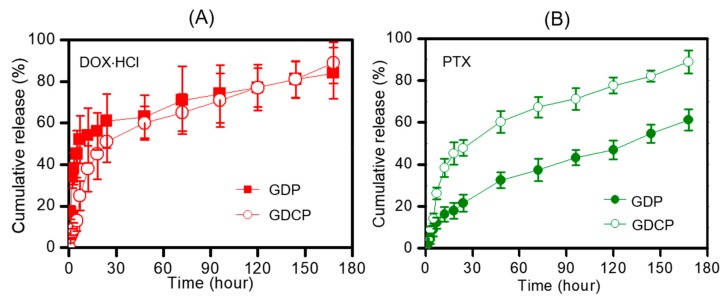
Release behaviors of (**A**) DOX⋅HCl and (**B**) PTX in GDP and GDCP hydrogels analyzed for 180 hours. These experiments were carried out three times at 37 °C under 100 rpm. These values were expressed as the mean ± standard deviation.

**Figure 3 ijms-20-04671-f003:**
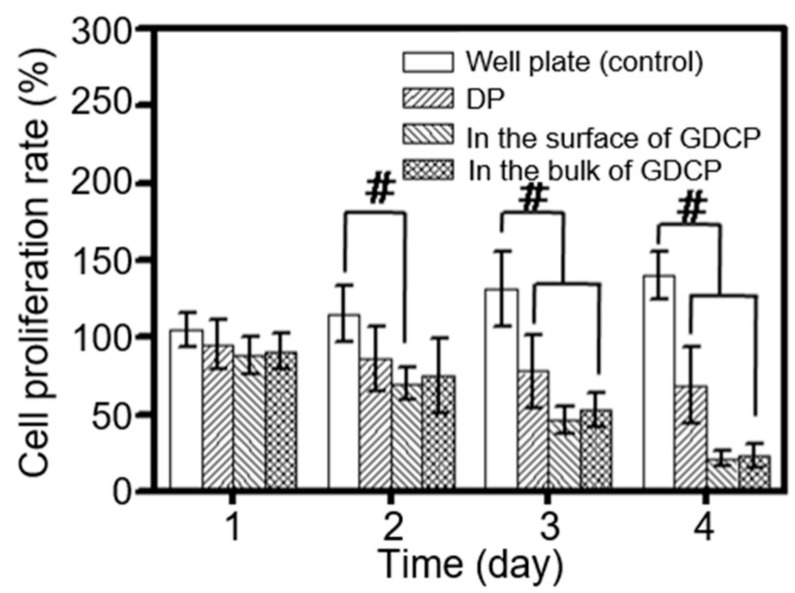
In vitro cell proliferation rates of MCF-7 breast cancer cells cultured on control, DP, and GDCP at 37 °C for 1, 3, 5, and 7 days. This experiment was carried out three times (^#^*p* < 0.05). These values were expressed as the mean ± standard deviation.

**Figure 4 ijms-20-04671-f004:**
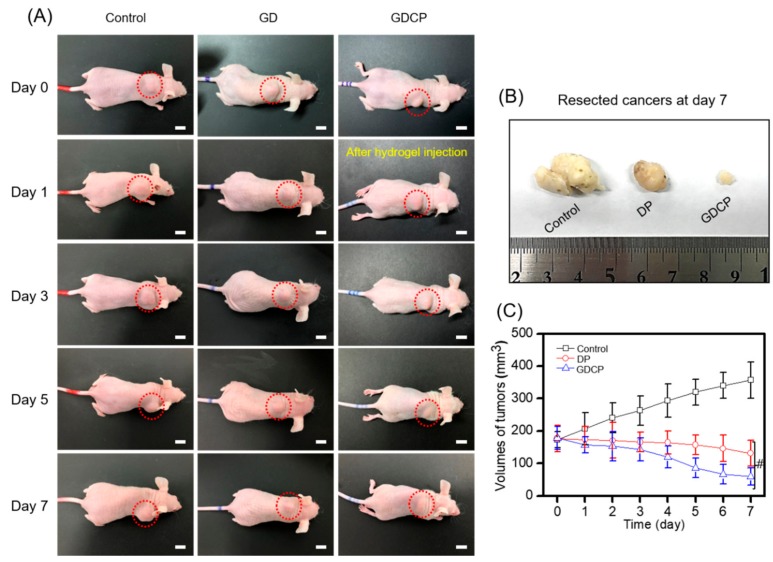
(**A**) Gross appearances of cancers bearing to the backs of control and DOX⋅HCl/PTX- and GDCP-treated mice observed before intravenous injection, and at Day 1, 3, 5, and 7 after the injection. (**B**) Cancer resected from control and DOX⋅HCl/PTX- and GDCP-treated mice at Day 7. (**C**) Volumes of tumors in control and DOX⋅HCl/PTX- and GDCP-treated mice (^#^*p* < 0.05). These values were expressed as the mean ± standard deviation. Scale bar indicates 1 cm.

**Figure 5 ijms-20-04671-f005:**
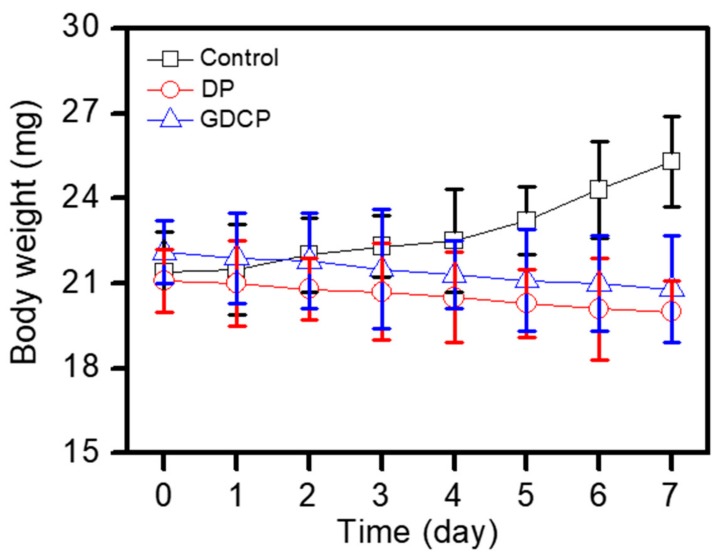
Body weights (mg) of control and DP- and GDCP-treated mice measured before intravenous injection, and at Day 1, 3, 5, and 7 after the injection. This experiment was carried out three times.

**Figure 6 ijms-20-04671-f006:**
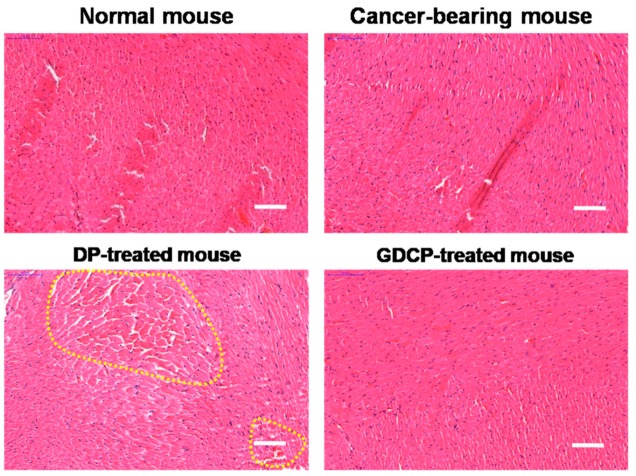
H&E-stained slides of hearts extracted at Day 7 from normal, cancer-bearing, and DP and GDCP-treated mice. Yellow dotted lines indicate abnormal cardiotoxicities including disorganization of the myocardium and myofibrillar fragmentation. Scale bar indicates 100 μm.

**Figure 7 ijms-20-04671-f007:**
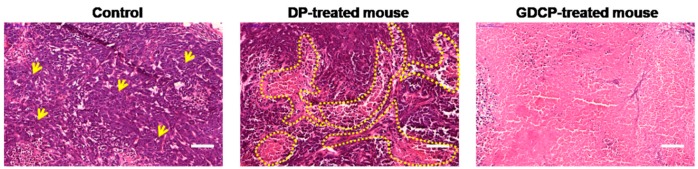
H&E-stained slides of cancers extracted at Day 7 from normal, cancer-bearing, and GDCP hydrogel-treated mice. In the control, clustered cancer cells were generally observed; on the other hand, in the DP-treated mouse, clustered cancer cells and a necrotic area were partially observed. Overall, necrotic area was observed in GDCP-treated mouse. Yellow arrows indicate typical clustered cancer cells. Yellow dotted lines indicate necrotic area. Scale bar indicates 100 μm.

**Figure 8 ijms-20-04671-f008:**
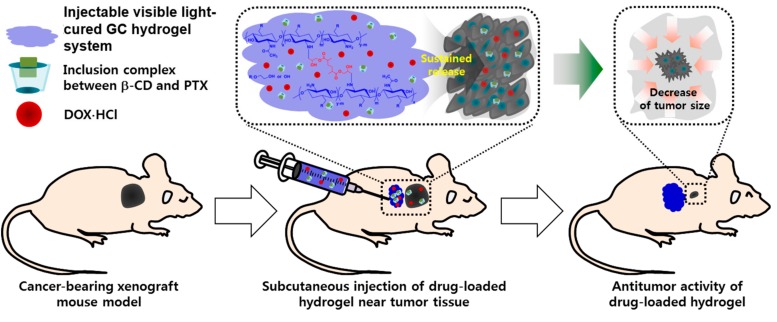
Expected mechanism of GDCP on the anticancer effect of breast cancer in vivo.

**Table 1 ijms-20-04671-t001:** Samples for this study.

Abbreviation	Explanation
GDCP	DOX⋅HCl/PTX-complex β-CD-loaded GC hydrogel
GDP	DOX⋅HCl/PTX-loaded GC hydrogel
DP	DOX⋅HCl/PTX

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
