# Peer review of "Hydrogel-Mediated DOX⋅HCl/PTX Delivery System for Breast Cancer Therapy"

_ijms, 2019, doi:10.3390/ijms20194671_

Round 1

Reviewer 1 Report

In this study the authors developed a chitosan-based hydrogel drug delivery system for breast cancer therapy. The article delivers some useful information and certainly add to the body of the literature. Nevertheless, my main concern is how the data are presented and how the work has been done. The materials and method section and in particular the presentation and translation of the data is not clear. The statistical analysis is very confusing and unclear. Example is Figure 3.

The section 4.10, the statistical analysis need clearly identify the data sets that have been evaluated.

In addition, there is not enough information on how the histological analysis was performed and it is hard to correlate the provided result and discussion in section 2.6 and 2.7 to the provided histological analysis.

The text also suffers from the long abbreviations which make it hard to follow, and I suggest the authors to provide a table explaining different terms,

In section 4.8. what authors mean by 2mg? this can not be a concentration and in addition it is not also clear if they have considered a control hydrogel which only contains cells.

Given the poor materials and methods and weak interpretation and presentation of data I can not suggest this paper for publication in IJMS in its current form. This paper can be considered after major revision.

Author Response

Dear Reviewer #1,

Thank you for evaluation of the manuscript entitled “Hydrogel-mediated DOX×HCl/PTX delivery system for breast cancer therapy”. We greatly appreciate the reviewer to give us the valuable comments. We agreed their suggestions; therefore, the manuscript was revised manuscript covers ever point of the suggestions and/or recommendations from the reviewer. Also, one co-author, Prof. Heung Jae Chun, who contributes to the revision was added in this manuscript. The details of revisions are as follows.

Reviwer #1

In this study the authors developed a chitosan-based hydrogel drug delivery system for breast cancer therapy. The article delivers some useful information and certainly adds to the body of the literature. Nevertheless, my main concern is how the data are presented and how the work has been done. The materials and method section and in particular the presentation and translation of the data is not clear. The statistical analysis is very confusing and unclear. Example is Figure 3.

Q1) The section 4.10, the statistical analysis need clearly identify the data sets that have been evaluated.

A1) We appreciate your valuable comment. The section was revised as you noted. The statistical analysis was carried out about data of in vitro cell viability, cancer volume and body weight; therefore, the information was expressed in Section 4.10.

Q2) In addition, there is not enough information on how the histological analysis was performed and it is hard to correlate the provided result and discussion in section 2.6 and 2.7 to the provided histological analysis.

A2) We appreciate your valuable comment. The histological results were evaluated by two observers from different agencies. The information was in detail added in Section 4.9. Histological evaluation of cancer/heart tissues can be supported by some references as bellows:

Cancer tissue:

Scientific Reports, 2016, 6, 21225.

Heart tissue:

Int. J. Nanomedicine, 2017, 12, 7103-7119.

Anatol. J. Cardiol. 2016, 16, 234-241.

Q3) The text also suffers from the long abbreviations which make it hard to follow, and I suggest the authors to provide a table explaining different terms.

A3) We appreciate your valuable comment. The authors revised the abbreviations of samples, and the explanation was shown in Table 1.

Q4) In section 4.8. what authors mean by 2mg? this can not be a concentration and in addition it is not also clear if they have considered a control hydrogel which only contains cells.

A4) We appreciate your valuable comment. The administration amount of anticancer drugs is also expressed as mg/kg (weight of mouse) as bellow references; in this study, to adjust the concentration of anticancer drug, administration amount had better to express as mg/kg.

1) Int. J. Nanomedicine, 2018, 13, 1723-1736.

2) J. Mater. Chem. B, 2015, 3, 1518-1528.

Q5) Given the poor materials and methods and weak interpretation and presentation of data I can not suggest this paper for publication in IJMS in its current form. This paper can be considered after major revision.

A5) We appreciate your valuable comment. The authors agree to the reviewer’s comment; therefore, the methods were in detail revised as you noted.

Reviewer 2 Report

The work presented in this manuscript aims to draw attention to the limitations with current chemotherapy delivery systems for the treatment of breast cancer. Hydrogel-mediated drug delivery systems have shown promise to overcome these limitations and using an injectable format that can be translated to the clinic is presented as a viable option. Although the authors should be commended for the large amount of work that has gone into the development of this article, I believe in its current form it does not meet the standard for the International journal of molecular sciences.

Work should be undertaken to raise the standard of grammar as there are numerous grammatical and sentence structural issues throughout the manuscript. See below for details. There is also a number of issues with the experimental design of the in vitro study. Adequate controls to assess the impact of seeding cells within a hydrogel-alone are not shown. These effects between cells grown on a plastic monolayer and cells embedded within a hydrogel are are not established. There is also a treatment group of encapsulated paclitaxel without hydrogel that is not included. I also have concerns about some of the conclusions reached from the limited study performed. I believe efforts to quantify histological differences between treatment groups would increase the confidence in the claims made. The introduction could also be expanded to cover not only more detailed background of drug delivery systems but also breast cancer and current state of he art hydrogel delivery systems. A much more detailed methods section would also establish greater confidence in the study.    

Minor revisions

Line 29: ‘worldwide’

Line 42: ‘delivery instead of discovery’

Line 44: In addition to systemic drug delivery ‘complications’

Line 68: Please indicate if error bars are SD or SEM

Line 86: compared with cells cultured in a (96, 12?) well-plate

Line 100 When DOX-HCl/PTX was locally delivered and GC/DOX-HCl/CD/PTX was systematically – is the sentence worded correctly?

Line 115: Volumes of tumors

Line 267: injected close to the tumor.

Line 275: observed by using fluorescence microscopy? Yet no fluorescence images are shown only colour H&E are in the figure?

Major Comments

Figure 3.

The figure is difficult to interpret. Is cell viability normalised at each timepoint? If not MC-7 cells would be expected to proliferate much more than 50% over 7 days. Have the cells been seeded at a density to avoid reaching confluency before the 7 day end point of the assay? If normalised, variability seems to be very high for an in vitro viability assay. At the drug concentrations used much greater cell death should have occurred with MCF-7 cells over 7 days. Wouldn’t statistical analysis between time points of different treatment conditions be more relevant for the comparison than between timepoints of the same treatment group? I believe there should be another treatment group included here - the cyclodextrin encapsulated paclitaxel not within the hydrogel. As any differences observed may be as a result of increased water solubility of Paclitaxel and not an effect of the release rate of the hydrogel.

Line 88: ‘because PTX is in a more water-soluble form than in the latter.’ Can this claim really be made from the data provided, I don’ think there is confidence in this statement.

Line 128: There is no quantification from the histological analysis or even annotation indicating the abnormal cells and histology. Details of how this histological assessment was made may provide greater confidence in the conclusions reached.   

Materials & Methods

Details of how exactly the hydrogel was injected and cured in vivo have not been fully explained

Should the paclitaxel not have been diluted in another vehicle besides water due to its poor solubility? As this is not clinically relevant why was water used for the in vivo studies as a comparison to a systematic treatment?

For the in vitro assay there should be a treatment control group that is cells in hydrogel only (not just compared to monolayer conditions).

Drugs are not recorded as concentrations (i.e. 2mg). This does not indicate concentrations within a well, hey should be listed as mg/ml or molarity.   

Author Response

Dear Reviewer #2,

Thank you for evaluation of the manuscript entitled “Hydrogel-mediated DOX×HCl/PTX delivery system for breast cancer therapy”. We greatly appreciate the reviewer to give us the valuable comments. We agreed their suggestions; therefore, the manuscript was revised manuscript covers ever point of the suggestions and/or recommendations from the reviewer. Also, one co-author, Prof. Heung Jae Chun, who contributes to the revision was added in this manuscript. The details of revisions are as follows.

Reviewer #2,

The work presented in this manuscript aims to draw attention to the limitations with current chemotherapy delivery systems for the treatment of breast cancer. Hydrogel-mediated drug delivery systems have shown promise to overcome these limitations and using an injectable format that can be translated to the clinic is presented as a viable option. Although the authors should be commended for the large amount of work that has gone into the development of this article, I believe in its current form it does not meet the standard for the International journal of molecular sciences.

Work should be undertaken to raise the standard of grammar as there are numerous grammatical and sentence structural issues throughout the manuscript. See below for details. There is also a number of issues with the experimental design of the in vitro study. Adequate controls to assess the impact of seeding cells within a hydrogel-alone are not shown. These effects between cells grown on a plastic monolayer and cells embedded within a hydrogel are are not established. There is also a treatment group of encapsulated paclitaxel without hydrogel that is not included. I also have concerns about some of the conclusions reached from the limited study performed. I believe efforts to quantify histological differences between treatment groups would increase the confidence in the claims made. The introduction could also be expanded to cover not only more detailed background of drug delivery systems but also breast cancer and current state of he art hydrogel delivery systems. A much more detailed methods section would also establish greater confidence in the study.    

Minor revisions

Q1) Line 29: ‘worldwide’

A1) It was revised as you noted.

Q2) Line 42: ‘delivery instead of discovery’

A2) It was revised as you noted.

Q3) Line 44: In addition to systemic drug delivery ‘complications’

A3) It was revised as you noted.

Q4) Line 68: Please indicate if error bars are SD or SEM

A3) It was revised as you noted.

Q5) Line 86: compared with cells cultured in a (96, 12?) well-plate

A5) It was revised as you noted.

Q6) Line 100 When DOX-HCl/PTX was locally delivered and GC/DOX-HCl/CD/PTX was systematically – is the sentence worded correctly?

A6) It was revised as you noted.

Q7) Line 115: Volumes of tumors

A7) It was revised as you noted.

Q8) Line 267: injected close to the tumor.

A8) It was revised as you noted.

Q9) Line 275: observed by using fluorescence microscopy? Yet no fluorescence images are shown only colour H&E are in the figure?

A9) Section of histological evaluation was rewritten in detail and used observation method was revised.

Major Comments

Q10) Figure 3.

The figure is difficult to interpret. Is cell viability normalised at each timepoint? If not MC-7 cells would be expected to proliferate much more than 50% over 7 days. Have the cells been seeded at a density to avoid reaching confluency before the 7 day end point of the assay? If normalised, variability seems to be very high for an in vitro viability assay. At the drug concentrations used much greater cell death should have occurred with MCF-7 cells over 7 days. Wouldn’t statistical analysis between time points of different treatment conditions be more relevant for the comparison than between timepoints of the same treatment group? I believe there should be another treatment group included here - the cyclodextrin encapsulated paclitaxel not within the hydrogel. As any differences observed may be as a result of increased water solubility of Paclitaxel and not an effect of the release rate of the hydrogel.

Q11) Line 88: ‘because PTX is in a more water-soluble form than in the latter.’ Can this claim really be made from the data provided, I don’ think there is confidence in this statement.

A11) We appreciate your valuable comment and agree to your notice. The authors deleted the sentence because the reviewer’s thought is reasonable.

Q12) Line 128: There is no quantification from the histological analysis or even annotation indicating the abnormal cells and histology. Details of how this histological assessment was made may provide greater confidence in the conclusions reached.   

A12) We appreciate your valuable comment. The histological results were evaluated by two observers from different agencies. The information was in detail added in Section 4.9. Histological evaluation of cancer/heart tissues can be supported by some references as bellows:

Cancer tissue:

Scientific Reports, 2016, 6, 21225.

Heart tissue:

Int. J. Nanomedicine, 2017, 12, 7103-7119.

Anatol. J. Cardiol. 2016, 16, 234-241.

Materials & Methods

Q13) Details of how exactly the hydrogel was injected and cured in vivo have not been fully explained

A13) We appreciate your valuable comment. The authors revised Section 4.9 in detail.

Q14) Should the paclitaxel not have been diluted in another vehicle besides water due to its poor solubility? As this is not clinically relevant why was water used for the in vivo studies as a comparison to a systematic treatment?

A14) We appreciate your valuable comment. This study evaluates the feasibility of DOX×HCl and PTX-loaded injectable GC hydrogel on breast cancer treatment for further clinical use; therefore, the authors thought that the co-administration of DOX×HCl and PTX. If PTX drugs are encapsulated in another vehicle, it may improve the tumor accumulation by passive targeting. This phenomenon may be not corresponded to our research concept. So, in in vivo animal test, PTX was dissolved in 0.01% DMSO. This method is well used for in vivo animal test.

Round 2

Reviewer 1 Report

the authors responded to my comments and in my opinion the paper can be accepted.